# Microgeographic differentiation in thermal and antipredator responses and their carry-over effects across life stages in a damselfly

**Nermeen R. Amer** [1,2] *, **Robby Stoks**[3], **Andrzej Antoł**[1], **Szymon Sniegula**[1]

**1** Department of Biodiversity, Institute of Nature Conservation, Polish Academy of Sciences, Krakow, Poland, **2** Department of Entomology, Faculty of Science, Cairo University, Giza, Egypt, **3** Department of Evolutionary Stress Ecology and Ecotoxicology, University of Leuven, Leuven, Belgium

* nraffatamer@gmail.com

**Data Availability Statement:** Here are our Raw data DOI and URL: DOI:10.5281/zenodo.10225341 https://zenodo.org/records/10225341.

## Abstract

Global warming and invasive species, separately or combined, can impose a large impact on the condition of native species. However, we know relatively little about how these two factors, individually and in combination, shape phenotypes in ectotherms across life stages and how this can differ between populations. We investigated the non-consumptive predator effects (NCEs) imposed by native (perch) and invasive (signal crayfish) predators experienced only during the egg stage or during both the egg and larval stages in combination with warming on adult life history traits of the damselfly *Ischnura elegans*. To explore microgeographic differentiation, we compared two nearby populations differing in thermal conditions and predator history. In the absence of predator cues, warming positively affected damselfly survival, possibly because the warmer temperature was closer to the optimal temperature. In the presence of predator cues, warming decreased survival, indicating a synergistic effect of these two variables on survival. In one population, predator cues from perch led to increased survival, especially under the current temperature, likely because of predator stress acclimation phenomena. While warming decreased, predator cues increased larval development time with a proportionally stronger effect of signal crayfish cues experienced during the egg stage, indicating a negative carry-over effect from egg to larva. Warming and predator cues increased mass at emergence, with the predator effect driven mainly by exposure to signal crayfish cues during the egg stage, indicating a positive carry-over effect from egg to adult. Notably, warming and predator effects were not consistent across the two studied populations, suggesting a phenotypic signal of adaptation at a microgeographic scale to thermal conditions and predator history. We also observed pronounced shifts during ontogeny from synergistic (egg and early larval stage) toward additive (late larval stage up to emergence) effects between warming and predator stress. The results point out that population- and life-stage-specific responses in life-history traits to NCEs are needed to predict fitness consequences of exposure to native and invasive predators and warming in prey at a microgeographic scale.

**Funding:** 1. Narodowe Centrum Nauki, Grant/ Award Number: 2019/33/B/NZ8/00521 2. Norwegian Financial Mechanism 2014-2021, Grant/Award Number: 2019/34/H/NZ8/00683 (ECOPOND) 3. Institute of Nature Conservation Polish Academy of Sciences The funders had no role in study design, data collection, and analysis, decision to publish, or preparation of the manuscript.

**Competing interests:** The authors have declared that no competing interests exist.

## Introduction

Organisms are typically exposed to multiple stressors which may interact so, it is important to study the individual and combined effects of these stressors to accurately assess their impact [1–4]. Two major stressors mediated by human activities, the introduction of invasive predators and warming can drastically affect native prey communities and whole ecosystems. For example, the introduction of an invasive alien predator might dramatically reduce native prey communities, as the later have not evolved recognition ability and hence defense mechanisms against new predators [5–7], supporting the so-called naïve prey hypothesis [5, 8, 9]. Warming may alter ectotherms' development rates, resulting in changed phenological events such as advanced hatching and emergence dates [10–12], which may, in turn, affect predator-prey interactions [8, 13–16].

In many animals with a complex life cycle, i.e. which go through discrete egg, larva, and adult stages, the degree of predation risk and the antipredator responses may differ between life stages [17–19]. For instance, the presence of non-consumptive predator cues of *Perca fluviatilis* fish during the egg stage decreased survival, but only up to two weeks after larval hatching, while the same exposure during the larval stage decreased survival and increased larval development time [17]. Moreover, predation risk imposed during a particular ontogenetic stage can be carried over to the following stage(s). This was shown in different invertebrate groups, including insects [20–22]. For example, exposure of freshwater shrimps to predation risk during the embryonic stage led to changes in larval morphology and development time [21]. Identifying and exploring carry-over effects is essential to evaluate the total impact of predation risk on prey performance.

Prey responses to predation risk and warming can differ between populations, even at a microgeographic scale [23–26]. It has been suggested that population-specific responses might be driven by habitat-specific exposure history to these potential stressors [24, 27, 28]. For example, thermal performance in Odonate larvae differed considerably between urban and adjacent rural sites [23, 29]. The mosquito larvae *Culex restuans* showed stronger antipredator responses to salamander kairomones and mosquito alarm cues in a population with increased predator density [26]. The ability of organisms to survive and grow under global warming not only depends on the physiological adjustments [30] but also on the ability of organisms to deal with the changes resulting from the interaction between temperature and predation risk [31]. For example, anti-predatory responses may change under warming [31]. This asks for cross-population studies that combine the effects of predation and warming-related stress in order to predict organismal performance under human-induced global change.

Here, we tested whether predation risk imposed throughout the egg stage only (acute exposition) or throughout the egg and larval stages (chronic exposition) affected prey traits at adult emergence through carry-over effects, and to what extent this can be modulated by warming in a semiaquatic insect, the damselfly *Ischnura elegans*. While similar studies on *I. elegans* exist [29, 32], our current research is novel because it examines the interactive effects of two important ecological stressors; global warming and biological invasion, throughout various life stages in prey (egg and larva). Additionally, our study focuses on different response patterns between populations at a micro-geographical scale. We compared the responses to cues from a native predator, perch fish (*Perca fluviatilis*) versus those from an invasive alien predator, the signal crayfish (*Pacifastacus leniusculus*). Along with the naïve prey hypothesis [8], we expect that the damselfly will only respond to native predator cues with lower survival, delayed egg hatching and adult emergence, and decreased mass at emergence. We expect the predator-induced carry-over effects to the adult stage to be higher under chronic exposure to the predator cues. Finally, we expect warming will increase the development rate, decrease survival and body

mass at emergence, and lower the predator-induced carry-over effect to the adult stage since higher temperature shortens the exposure time to predator cues.

## Material and methods

### Target species and study area

We chose *I. elegans* because it is a very common damselfly occurring from mid-Scandinavia and the United Kingdom to southern Italy and southern Spain [33], and it is neither protected nor endangered. In central Europe, the damselfly has one or two generations per year [34] with diapausing larvae as the overwintering stage [35]. Larvae hatch a couple of weeks after egg laying. Aquatic egg and larval stages commonly share habitats with top predators such as fish and crayfish [17, 36–38]. Moreover, *I. elegans* larvae occupy an intermediate position in aquatic food webs. They are significant predators of many aquatic invertebrates, including midges and mosquito larvae, while being prey for larger predatory invertebrates and fish [18]. It was shown that predator cues affect *I. elegans* damselfly life history and physiology during both the egg stage and larval stage [13, 17, 39].

The study animals were collected from two nearby ponds, Dąbski and Płaszowski, within the city of Kraków, southern Poland. No permits are required for damselfly sampling. Both ponds are situated within the Wisła river valley and are separated by a distance of 2.7 km. Based on the short distance, and previously published literature [40], we assume that gene flow between these two populations is high. The ponds differ considerably in size, depth, thermal conditions, history of native and invasive crayfish occurrence, and other biotic and abiotic characteristics, which are described in detail in S4 Table. Based on pond characteristics and temperature logger reads, Dąbski pond holds on average colder water than Płaszowski pond (S2 Fig, S5 Table). Dąbski pond was reported to obtain one native crayfish, the noble crayfish (*Astacus astacus*), and two invasive alien crayfish species, spiny-chick crayfish (*Faxonius limosus*) and swamp crayfish (*Procambarus clarkia*). Instead, Płaszowski pond contains only one native crayfish species, Danube crayfish (*Astacus leptodactylus*). We used cues from two different predator species to induce non-consumptive predator effects (NCE). The first type of cues came from the native European perch (*P. fluviatilis*) which is a common fish species in Europe [41]. Both *I. elegans* sampling ponds hold perch populations (S4 Table). The second type of predator cues came from the invasive alien signal crayfish (*P. leniusculus*) that does not co-occur with *I. elegans* in the sampling ponds (S4 Table). Signal crayfish is native to North America. The crayfish was introduced to Europe, and Scandinavia specifically, in the 1960s. In the following years, the species was introduced to other European countries, including Poland. Currently, it is the most spread non-native crayfish on the continent and has a negative impact on local biodiversity [42]. The distance between the crayfish population and the damselfly collection ponds is ca. 200 km (Maciej Bonk and Rafał Maciaszek, pers. communication).

### Collection and housing of *I. elegans*

Adult females *I. elegans* in copulation were collected from the two ponds on 5 July 2020 using a standard method [43]. Females were placed in plastic containers with wet filter paper for egg laying. Containers with females were placed in a Styrofoam box with frozen cartridges to keep the temperature low. Females were transported by car to the Institute of Nature Conservation, Polish Academy of Sciences (INC PAS) in Kraków, where the laboratory experiment was run. Females were kept in a room with a natural photoperiod and a temperature of ca. 22˚C until egg laying. Newly laid clutches were transferred to two incubators (Pol-Eko ST700). The temperatures at which eggs and larvae were reared were adjusted once a week to follow seasonal

changes of mean weekly temperatures in shallow water, which is the optimal habitat for damselfly larvae [35].

## Experimental setup

We set up two temperature treatments (current temperature and +4˚C warming) crossed with three predator cue treatments in the egg stage and five predator cue treatments in the larval stage (Fig 1). At hatching, the egg control group continued as a larval control group (CC). The egg perch cue group was divided into larval control (PC) and larval perch cue (PP) groups. The egg signal crayfish cue group was divided into larval control (SC) and larval signal crayfish cue (SS) groups (Fig 1).

The two temperature treatments were the current temperature treatment which mimicked the current temperature in the collection site and the warming treatment where the current temperature was increased by 4˚C to mimic the predicted temperature change by 2100 [44]. The current temperatures were extracted from the Lake Model Flake [45]; in previous studies [43, 46, 47] it was shown this gives a reliable estimate of the field-measured water temperatures in ponds inhabited by the study species [48]. Except for overwintering conditions (see below), we followed weekly seasonal changes of daylight (photoperiod), with the inclusion of the Civil twilight at the sampling latitude. During the simulated winter, the temperature was set constant at 6˚C in the current temperature treatment and at 10˚C in the warming treatment. The light during winter was switched off. For the details of the applied temperature and photoperiod regimes, we refer to S1 Fig and S1 Table.

We installed three predator cue treatment groups in the egg stage: control, perch cues and signal crayfish cues. At hatching, these gave rise to five larval predator cue treatment groups

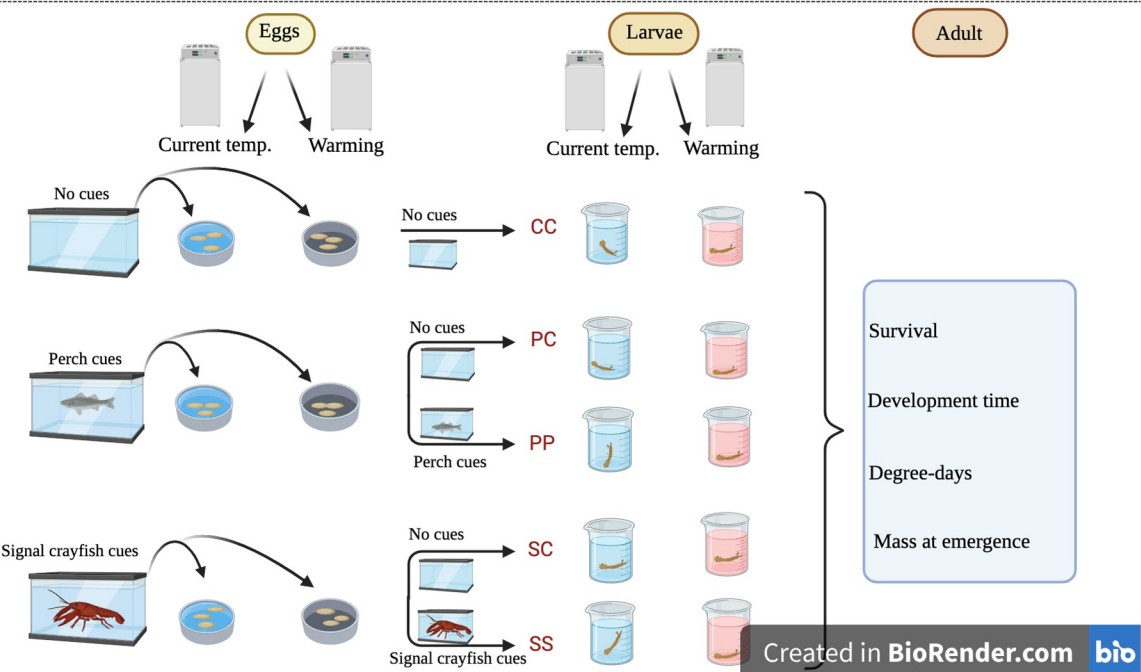

**Fig 1. Experimental design.** *Ischnura elegans* eggs were assigned to groups receiving no predator cues (control), native European perch cues (perch), or invasive alien signal crayfish cues (signal crayfish). At hatching, the egg control group continued as larval control group (CC). The egg perch cue group was divided into larval control (PC) and larval perch cue (PP) groups. The egg signal crayfish cue group was divided into larval control (SC) and larval signal crayfish cue (SS) groups.

(Fig 1). The control egg group continued as a control group in the larval stage. The two other predator cue treatment groups during the egg stage were each divided into two groups: a group that continued experiencing the same predator cues during the larval stage and a control group that did no longer exposed to predator cues during the larval stage. The latter group was designed to test for carry-over predator effect from the egg stage through the larval to the adult stages. Perch and signal crayfish cues were never combined.

To start, every egg clutch (each from a single female) was divided into six parts that were placed in separate plastic containers (15 x 11 x 7.5 cm) filled with 600 ml of water. In these containers, all clutches were pooled per population (Dąbski and Płaszowski pond). Water consisted of ¾ dechlorinated tap water and ¼ of dechlorinated tap water that, depending on the predator cue treatment contained predator cues originating from perch or signal crayfish. Eggs from every female (i.e., family) were present in every treatment combination during the egg stage: three predation cues (perch, signal crayfish, and control) crossed with two temperatures (current temperature and warming) treatments. Water refill took place every second day to keep the predator cue levels approximately constant [49]. At hatching, larvae were transferred to other containers (19 x 12 x 9 cm) filled with 1 l of water and kept in groups of 15 individuals (two group containers per treatment—a total of 30 individuals per treatment) for another 14 days. Group rearing during initial instars increases larval survival [50]. Next, larvae were individually placed in plastic cups (height 9 cm, diameter 4 cm), each filled with 100 ml of water. Water refill in cups holding individual larvae was analogous as in group-rearing containers (¾ dechlorinated tap water and ¼ of dechlorinated tap water with or without predator cues; refilled every second day). The larvae were fed twice a day (morning and afternoon) during weekdays and once a day during weekend days with laboratory-cultured *Artemia* sp. nauplii. For the first 14 days, larvae kept in groups received 10 ml of *Artemia* solution/container, while larvae kept in individual cups received 1 ml of an *Artemia* solution that contained on average 198.5 (SD = 92.4) nauplii/ml (N = 38 samples counted). During the simulated winter, larvae were fed once a day throughout the week. When larvae reached the pre-final instar (F-1), each individual was additionally provided with one living chironomid larva of a standardized size three times a week (Monday, Wednesday, and Friday).

## Collection and housing of the predators

The predators were collected a couple of weeks before we started the experiment. We collected perch from Dobczyce Lake in southern Poland (49˚52′18.316″N, 20˚2′30.937″E) and signal crayfish from Hańcza Lake in northern Poland (54˚15′9.522″N, 22˚48′36.86″E). Animals were transported to the INC PAS by car in aerated travel containers. In the laboratory, the predators were kept by species in aquaria with 52 L of dechlorinated and aerated tap water in the same cabinet at a constant temperature of 20˚C. The densities of predators in aquaria were based on the basal metabolic rate equations obtained for perch from [51] and for crayfish from [52]. To keep total metabolic rates balanced, the biomass of crayfish was made two times higher than the biomass of fish. After weighing, we kept two specimens of signal crayfish (wet mass 100 g) and two specimens of perch (wet mass 41.5 g) per experimental aquarium. We fed all predators frozen chironomid larvae (IT-Ichtyo Trophic, Stare Polichno, Poland) every second day (the larvae were thawed prior to feeding) and earthworms (myWORMS, Słupsk, Poland) once a week. Once a week, we changed 10 L of water in the predator aquaria. Perches were collected and housed with permission from the Local Ethical Committee (ref. 394/2020). Signal crayfish were collected with permission from the Regional Directorate for Environmental Protection in Białystok (ref. WPN.6205.21.2020.ML) and Nature Reserve Hańcza Lake and housed with

permission from the Regional Directorate for Environmental Protection in Kraków (ref. OP-I.672.8.2020.MK1).

## Response variables

Survival was measured as the number of larvae that survived until day 14 after hatching and until the day after adult emergence (hereafter, until emergence). Development time was measured as the number of days between oviposition and hatching (egg development time) and between hatching and emergence (larval development time). We also expressed development times in degree days (DDs), as DD better reflects the fact that insect development is strongly dependent on temperature. We calculated degree days by adding the average number of temperature degrees during the egg or larval stages. The lower temperature threshold for measuring DDs was 10˚C (the winter period in both temperature treatment groups was excluded when calculating DDs). We used this threshold because the minimum temperature of larval development in Odonata ranges between 8–12˚C [53]. Adult mass at emergence was measured as a wet mass one day after emergence. Animals were weighed to the nearest 0.1 mg using an electronic balance (Radwag® AS.62. R2 Plus).

Unfortunately, during the course of the experiment, we accidentally lost 27 individuals (S3 Table). Because of this, some treatment groups ended up with low sample sizes and were not considered in the analysis. Sample size per combination of temperature, predator cue treatment, and population ranged between 14 and 50 eggs, and between 3 and 22 larvae. There were two larval groups with a sample size of one individual (SC group from Dąbski population under warming, and CC group from Płaszowski pond under current temperature).

## Statistical analyses

For statistical analyses, we used R version 4.0.3 (R Core Team, 2013; RStudio Team, 2015). To test for survival, a generalized model (glm function) with binomial error distribution was used. For analyzing egg and larval development times in days and in DDs, and body mass, separate linear models (lm function) were used for each of the traits. Predator cues (control, perch, and signal crayfish), temperature (current and warming), population (Dąbski and Płaszowski), and sex were added as explanatory variables. Because the sex could not be determined at the start of the experiment, sex could not be added to the survival analysis. Initially, we ran full models that included all main effects and interaction terms. Subsequently, the interaction terms with $p > 0.05$ were removed from the final models, and sex, if not significant, was also removed from the final models to increase the power of the test.

## Results

Our statistical analyses showed significant main effects of predator cues, temperature, population, and sex, and in some cases two and three-way interaction effects between these variables on the life history of *I. elegans*.

### Egg development time

In general, predator cues increased, and warming decreased the egg development time expressed in days. Eggs took the longest time for their development when exposed to signal crayfish cues, and the shortest time in the control treatment without predator cues. Across all predator cue treatment groups, eggs from Dąbski pond took more time for their development than eggs from Płaszowski pond, but only under the current temperature treatment (predator cue × temperature × population interaction, Fig 2A, Table 1).

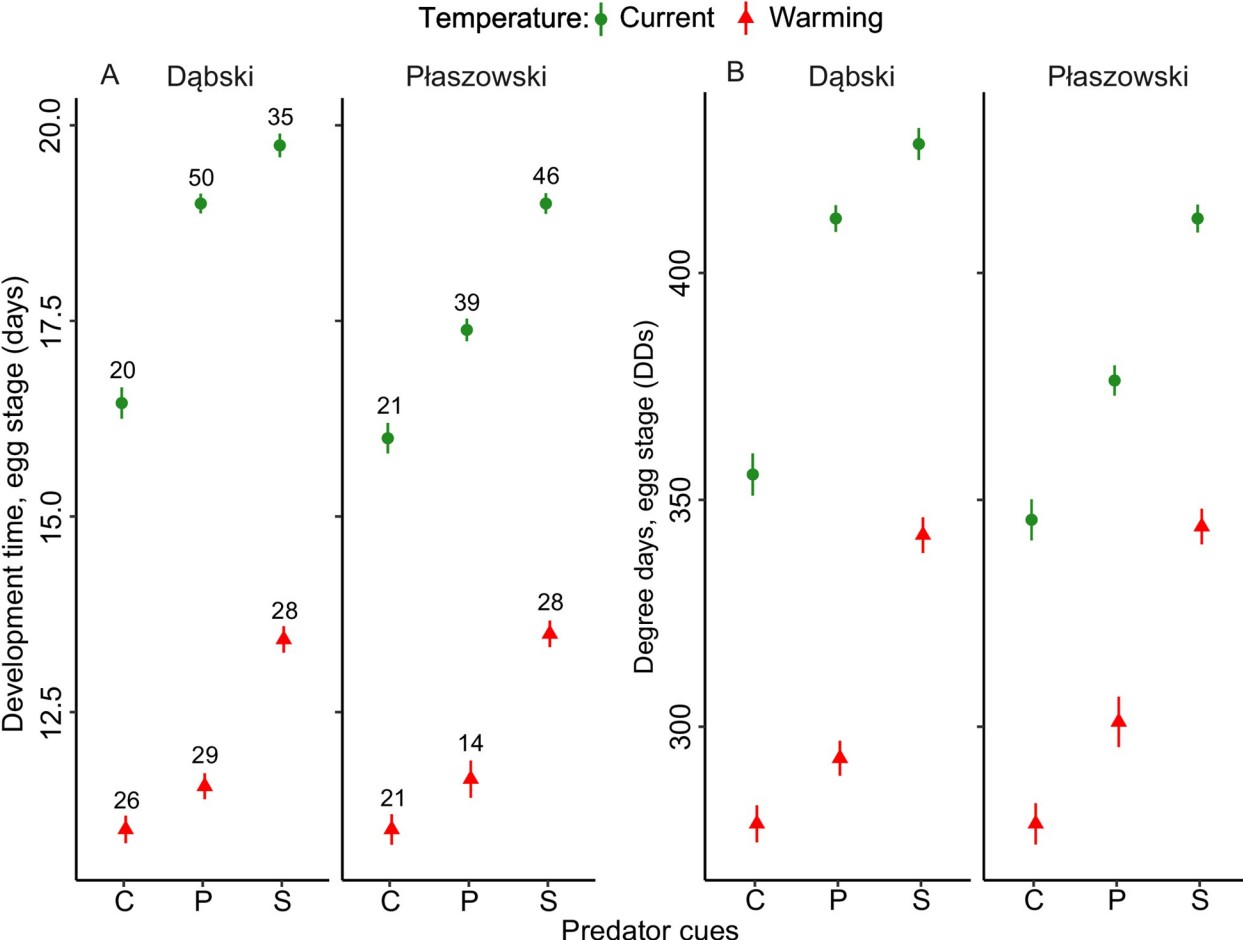

**Fig 2.** Effects of predator cues and temperature (current and 4°C warming) on the egg development time in days (A) and degree-days (B) in *I. elegans* from two populations (Dąbski and Płaszowski). Error bars show 95% CI. Sample sizes per group are provided in (A). Abbreviations: C–control; P- perch cues; S- signal crayfish cues.

Predator cues increased and warming decreased the egg development time expressed in degree-days (DDs). Eggs exposed to the signal crayfish cues required the most DDs, and eggs from the control group required the least DDs. Across all predator treatment groups, larvae from the Dąbski population needed more DDs than those from the Płaszowski population, but only under the current temperature treatment (Fig 2B, Table 1).

## Larval survival

Predator cues negatively affected larval survival until day 14 after hatching, but only under warming (interaction predator cue × temperature, Fig 3A, Table 2). Predator cues negatively affected survival in all groups, except for the chronic signal crayfish treatment at Płaszowski population (interaction predator cue × population Fig 3A, Table 2). Both acute and chronic perch exposure in the Płaszowski population and acute perch exposure in the Dąbski population caused the lowest survival rate (interaction predator cue × population, Fig 3A, Table 2).

Larval survival until emergence was positively affected by predator cues but only under current temperature (interaction predator cue × temperature). Warming positively affected survival only in the control treatment. Chronic exposure to perch cues led to increased survival,

**Table 1. Effects of predator cues, temperature, population, and their interactions on egg development time in days and in degree-days (DDs) in *I. elegans*.** Significant p-values are in bold.

| Predictor | df | Chisq | p-value |
| --- | --- | --- | --- |
| **Egg development time in days** | | | |
| Egg predator cue | 2 | 347.889 | **< 0.001** |
| Temperature | 1 | 1629.014 | **< 0.001** |
| Population | 1 | 10.064 | **0.002** |
| Predator cue x temperature | 2 | 71.196 | **< 0.001** |
| Predator cue x population | 2 | 30.483 | **< 0.001** |
| Temperature x population | 1 | 5.348 | **0.021** |
| Predator cue x temperature x population | 2 | 12.669 | **< 0.001** |
| **Egg development time in DDs** | | | |
| Egg predator cue | 2 | 314.139 | **< 0.001** |
| Temperature | 1 | 600.066 | **< 0.001** |
| Population | 1 | 9.156 | **0.003** |
| Predator cue x temperature | 2 | 67.980 | **< 0.001** |
| Predator cue x population | 2 | 27.427 | **< 0.001** |
| Temperature x population | 1 | 4.865 | **0.028** |
| Predator cue x temperature x population | 2 | 17.442 | **< 0.001** |

especially in Dąbski population reared under the current temperature (interaction predator cue × temperature x population; Fig 3B, Table 2).

## Larval development time

There was a trend (p = 0.076) that chronic exposure to perch and signal crayfish cues increased larval development time expressed in days. Warming decreased larval development time. Larvae of the Dąbski population required more days than those of the Płaszowski population for larval development. None of the interaction terms were significant (Fig 4A, Table 2).

Predator cues showed no significant effects on larval development expressed in DDs. Warming decreased the DDs. Larvae from the Dąbski population expressed more DDs in the larval stage than those from the Płaszowski pond. None of the interaction terms were significant (Fig 4B, Table 2).

## Mass at emergence

Predator cues significantly increased mass at emergence, and this pattern was mainly driven by the acute signal crayfish cue (SC) exposure (note that this group in the Dąbski population was excluded from the analysis due to low sample size), indicating a positive carry-over effect of signal crayfish cues from the egg until adult emergence (Fig 5, Table 2). After adding development time as a covariate, the increase in mass under predator stress was still significant (Chisq = 5.072, df = 4, p = 0.001), indicating that prolonged development under predator cues could not completely explain the mass increase. Warming increased mass at emergence. Individuals from the Dąbski population were heavier than those from the Płaszowski population. Females were heavier than males. No interaction terms were significant (Fig 5, Table 2).

## Discussion

Our results revealed population-specific life history responses to NCEs from native and invasive alien predators and to warming at a microgeographic scale. Another important finding

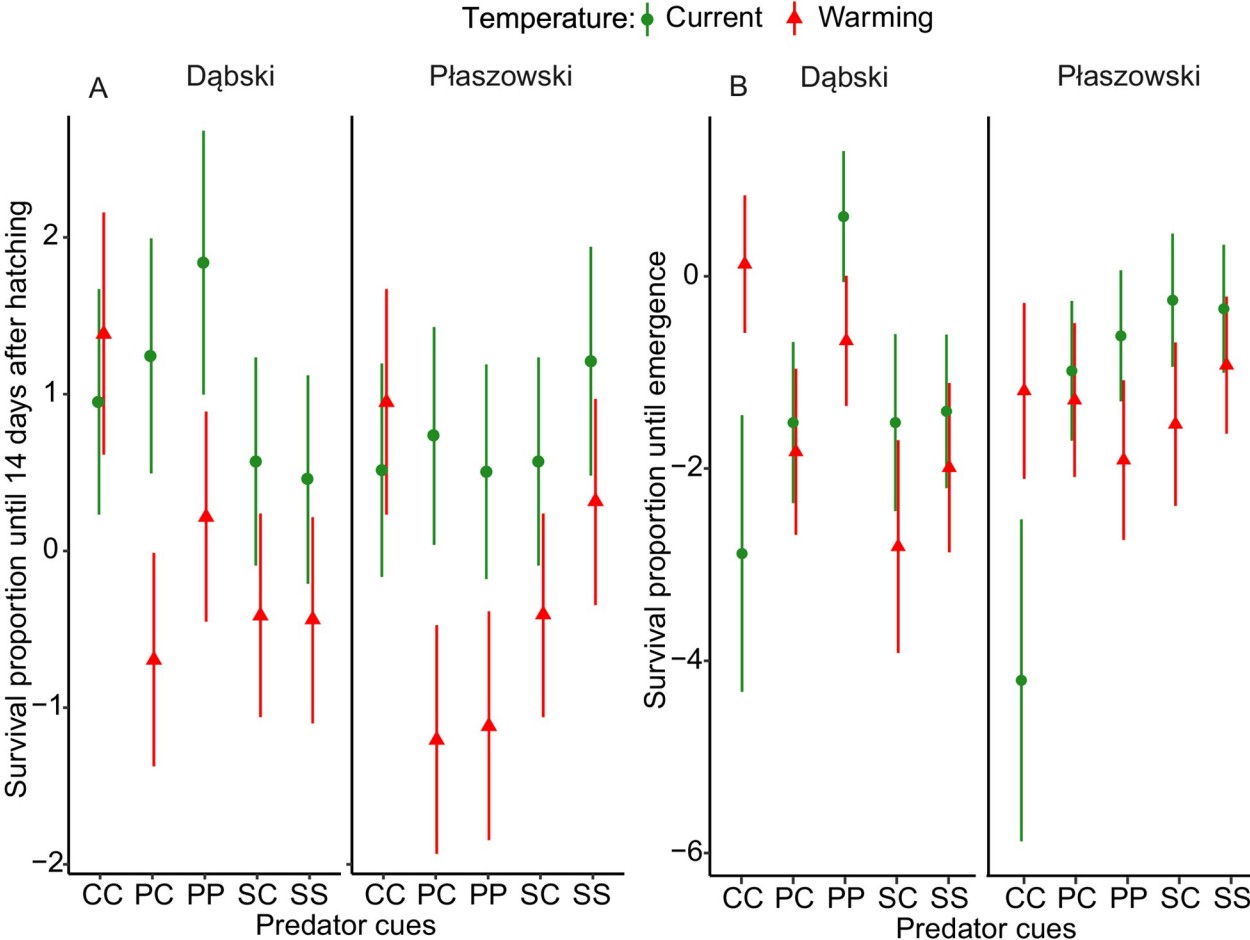

**Fig 3.** Effects of predator cues and temperature (current and 4˚C warming) on larval survival after 14 days (A) and until emergence (B) in *I. elegans* from two populations (Dąbski and Płaszowski). Error bars show 95% CI. Abbreviations: CC—control (egg) and control (larva) group; PC—perch (egg) and control (larva) group; PP—perch (egg) and perch (larva) group; SC—signal crayfish (egg) and control (larva) group; SS—signal crayfish (egg) and signal crayfish (larva) group.

was that NCEs strongly affected egg development time and also carried over to larval and adult stages. These results point out that a full life cycle approach is needed as even when a stressor is experienced during a brief initial egg stage it can shape adult fitness traits through carry-over effects.

Larval survival decreased under predation risk, but only under warming conditions and until 14 days after hatching. Similar to the current results, perch cues decreased larval survival of *I. elegans* during the first two weeks after hatching [17]. When no predator cues were present, the increase in temperature had a beneficial impact on damselfly survival until emergence, likely because the warmer conditions approached the optimal temperature range. However, when predator cues were introduced, warming had the opposite effect, leading to a decrease in survival. This suggests a synergistic relationship between these two variables in influencing survival. Interestingly, in one population (Dąbski), the presence of predator cues from perch actually improved survival, particularly under current temperature conditions. This result is in contrast to previous studies on damselflies [17, 54, 55]. We do not have a clear explanation for the increased survival until emergence under native predator stress. Nevertheless, it is plausible

**Table 2. Effects of predator cues, temperature, population on larval survival (14 days after hatching and until emergence), development times in days and in degree-days (DDs), and mass in *I. elegans* (sex included).** Significant p-values are in bold.

| Predictor | df | Chisq | p-value |
| --- | --- | --- | --- |
| **Larval survival 14 days after hatching** | | | |
| Predator cue | 4 | 6.210 | 0.184 |
| Temperature | 1 | 1.057 | 0.304 |
| Population | 1 | 1.057 | 0.304 |
| Predator cue x temperature | 4 | 18.724 | < **0.001** |
| Predator cue x population | 4 | 14.634 | **0.006** |
| **Larval survival until emergence** | | | |
| Predator cue | 4 | 11.210 | **0.024** |
| Temperature | 1 | 2.518 | 0.112 |
| Population | 1 | 0.062 | 0.802 |
| Predator cue x temperature | 4 | 41.723 | < **0.001** |
| Predator cue x population | 4 | 28.257 | < **0.001** |
| Predator cue x temperature x Population | 5 | 15.966 | **0.007** |
| **Larval development time in days** | | | |
| Predator cue | 4 | 2.225 | 0.069 |
| Temperature | 1 | 319.196 | < **0.001** |
| Population | 1 | 5.414 | **0.021** |
| **Larval development time in DDs** | | | |
| Predator cue | 4 | 1.945 | 0.105 |
| Temperature | 1 | 141.886 | < **0.001** |
| Population | 1 | 5.632 | **0.019** |
| **Mass** | | | |
| Predator cue | 4 | 3.031 | **0.020** |
| Temperature | 1 | 67.440 | < **0.001** |
| Population | 1 | 4.412 | **0.037** |
| Sex | 1 | 10.334 | **0.001** |

that the elevated survival until emergence could be attributed to acclimation to stress [4]; individuals that managed to survive under predator stress during the egg and early larval stages may have subsequently exhibited reduced responses after continued exposure to the same predator cues throughout the subsequent larval stages and up to emergence.

Separate analyses of NCE during the egg and larval stages showed that the difference in development time under predation pressure was driven mainly by egg responses to predator cues. This matches the idea that the immobile egg stage is the most sensitive phase to predation and further indicates that the negative predator-induced carry-over effect from the egg through the larval stage up to adult emergence decreases with exposure duration during the larval stage. Delayed development under predation pressure might be an effect of physiological stress in prey. Such stress could lead to re-allocation of energy to costly defense mechanisms instead of maintaining a fast development rate [12]. The effects of predator cues on development have been shown in previous studies. For example, *I. elegans* showed a longer egg development time in response to signal crayfish cues, which supports our results, but no response toward perch cues [39], which is in contrast to current results. Larvae briefly exposed to fish cues during initial larval instars decreased their growth rate in later instars [56]. Another study on confamiliar species showed either decreased (*Coenagrion pulchellum*) or increased (*Enallagma cyathigerum*) egg development times under predation risk [56]. These contrasting

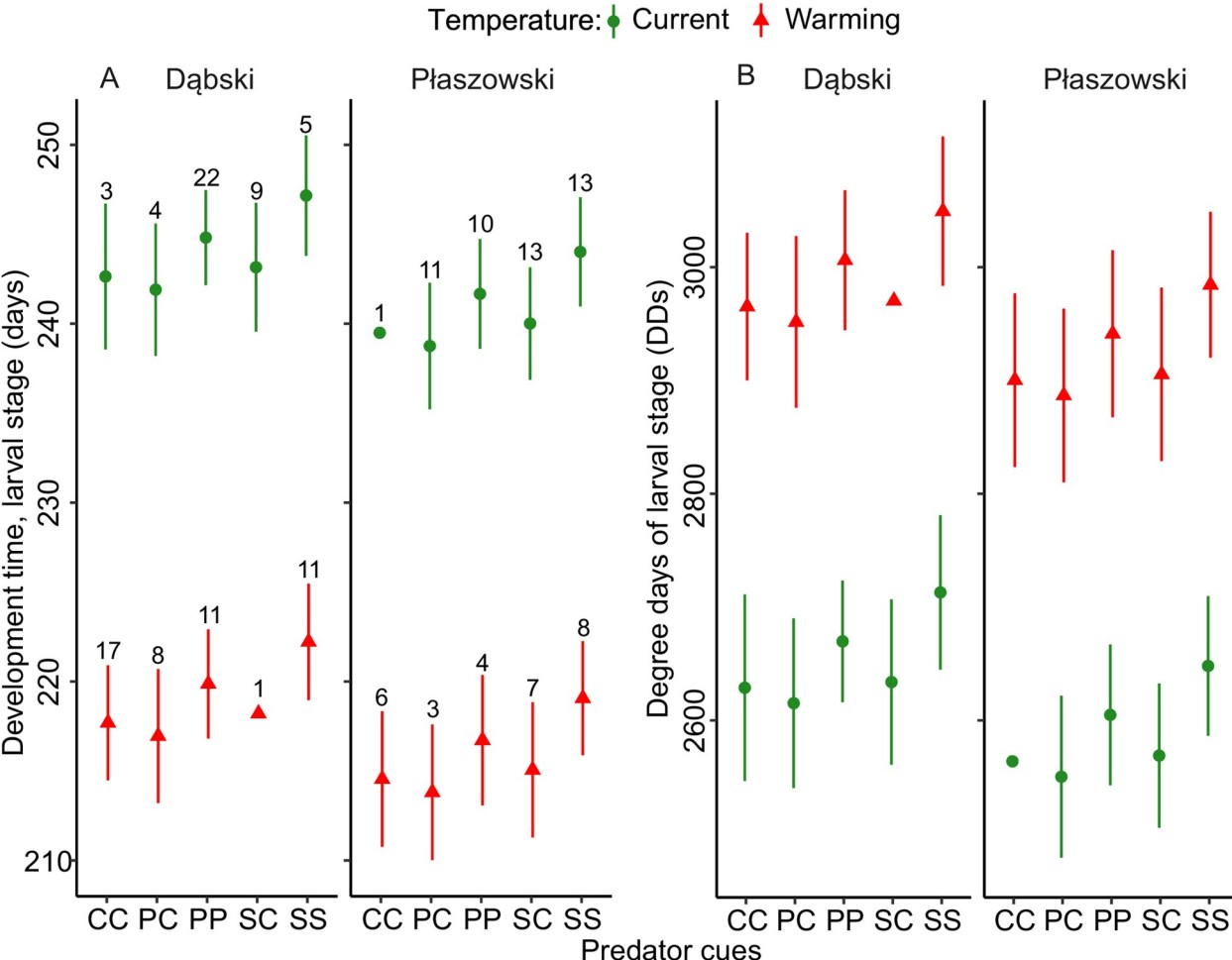

**Fig 4.** Effects of predator cues and temperature (current and warming) on the larval development time in days (A) and degree-days (B) in *I. elegans* from two populations (Dąbski and Płaszowski). Error bars show 95% CI. The sample size per group is provided in (A). Abbreviations as in Fig 3.

responses to predator cues might be due to population-specific experiences with different predator species in nature and different rearing methods, e.g., constant vs. changing temperatures and photoperiods. Conspecific populations from different habitats may respond differently when exposed to the same stressor. For example, in two independent studies on *I. elegans* eggs, one showed a significant effect of perch cues on egg development time [57], while the other study revealed no effect of perch cues on egg development time [39]. NCE on prey development might vary considerably between prey developmental stages, populations, and species, likely due to different defense systems used by prey [58].

A stronger delay in development under predator stress was found when individuals were exposed to the invasive crayfish cues than the native perch cues in the egg stage, and under chronic (egg and larval stages) than acute (egg stage only) exposition. The presence of an intense reaction to kairomones from the invasive crayfish might be explained by damselfly exposure to phylogenetically related crayfish species [59] that have been recorded in the two studied ponds (S4 Table). A stronger response under chronic exposure to predator cues could result from weak or no physiological reconfiguration in the prey, in which prey reduces the costs of physiological adjustments under risk conditions [60]. Although the pattern of

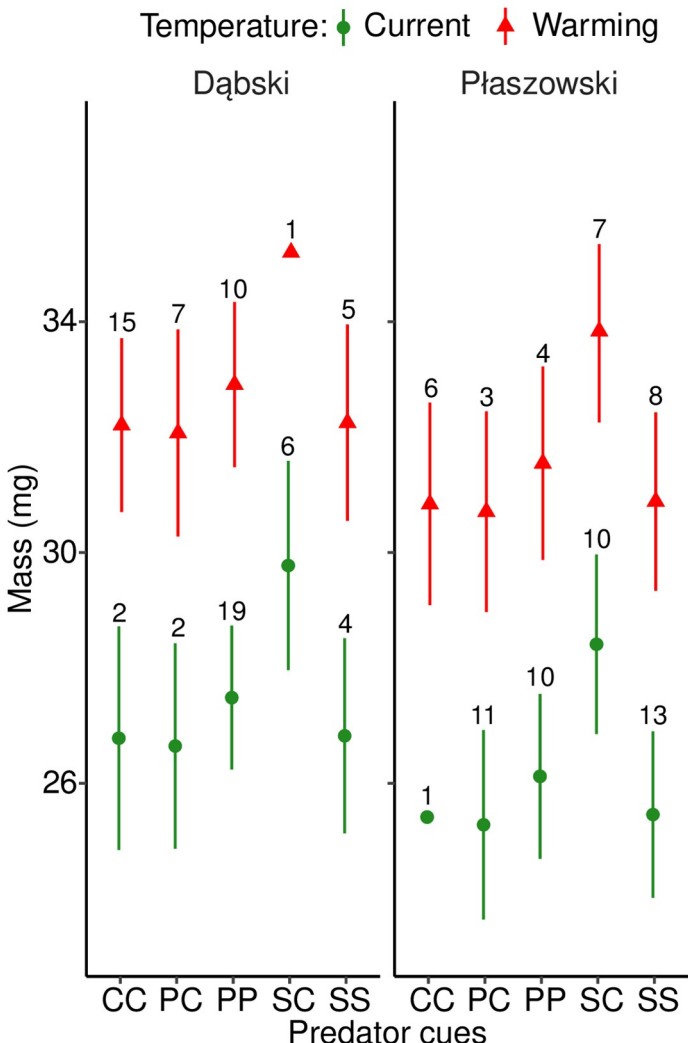

**Fig 5. Effects of predator cues and temperature (current and warming) on the estimated mass at emergence in *I. elegans* from two populations (Dąbski and Płaszowski).** Error bars show 95% CI. Sample size per group is provided. Abbreviations as in Fig 3.

responses to predator cues during the larval stage was similar across the two populations and temperatures, the response of the egg stage was not consistent between both populations. Specifically, eggs from the Dąbski pond took more time for their development under predator stress than eggs from the Płaszowski pond, but only at the current temperature. A higher sensitivity to predator cues by eggs from the Dąbski pond than from the Płaszowski pond might be a consequence of damselfly experience with, respectively; multiple and single crayfish species in nature (S4 Table). This suggestion is based on earlier studies showing a stronger effect of multiple predator species on prey responses to predation risk [26, 61–63]. For example, *Culex restuans* mosquitoes that experienced a high level of background predation risk based on pond-population predation history, showed increased wriggling activity in response to predation cues [26]. Warming reduced the NCE on egg development time likely due to faster development hence shorter exposure time under warming temperature as shown in the study of [64] and /or faster biodegradation of predator cues at the higher temperature, which was also

found in previous case studies [65]. Such reduced NCEs under warming also removed the difference between populations in the egg response to predator cues. In summary, the results imply that egg and larval stages show different sensitivity to predator stress in terms of development rate and that warming may decrease this sensitivity and lead to similar responses to native and invasive predators.

When we expressed development time in terms of thermal units, i.e. degree-days (DDs), required for development, we noted qualitatively the same results as for estimates of development time expressed in days (compare Figs 2A and 4A vs. Figs 2B and 4B). Because the experimental simulation of seasonal changes in temperature was equally experienced by all reared individuals, may suggest that patterns in development time in days and DDs reflected thermal phenotypic evolution in the two local populations. However, maternal effects may partially have contributed to phenotypic differences between populations [66]. Studies on the contribution of maternal effects in damselfly offspring life history traits indicated that a considerable proportion of variance in egg, but not larval, traits was driven by maternal effects [67, 68].

Damselflies from both populations showed an increased mass at emergence when exposed to predator cues, and this result was mainly driven by an acute egg exposure to signal crayfish, i.e. a positive carry-over effect from the egg to adult emergence. A higher mass under predation stress was not just a side-effect of prolonged development as the pattern remained the same after correcting mass by development time. The stronger effect of crayfish than perch cues suggests that damselfly eggs are more prone to stress caused by predation risk from crayfish than from perch. Juvenile and mature signal crayfish are omnivorous and feed on substrates into which *I. elegans* eggs might be oviposited, with faster crayfish growth if the substrate contains invertebrates [69]. In comparison, juvenile and mature perch feed on mobile prey such as zooplankton, invertebrates, and fish [70], hence the fish is less likely to prey upon damselfly eggs.

In both temperature treatments, damselflies from the warmer Płaszowski pond expressed a shorter development time in days and DDs until emergence than individuals from the colder Dąbski pond, and this is regardless of the predator cue treatment. Such thermal differentiation in development might reflect co-gradient variation where both genes and a warmer environment direct the population towards the fastest development [71], a pattern less often reported in ectotherms [72, 73]. In our case, this may reflect up-regulation of the expression of genes regulating development in the population from the warmer pond and down-regulation of these genes in the population from the colder pond. Such a pattern of up-regulated genes in individuals reared under elevated temperature was shown in *I. elegans* larvae from several other populations in southern Poland [74]. Furthermore, as a warming climate leads to an extension of the growth season, this can lead to an increased number of generations per growth season, i.e. from univoltine to bivoltine [34]. Note that the thermal differentiation between populations appeared despite a strong homogenization effect of gene flow at a microgeographic scale, as was indicated in previous population genetic studies on mobile ectotherms, including *I. elegans* [25, 75, 76].

Experimental warming shortened the development time of individuals from both populations and increased mass at emergence. In ectotherms, increasing temperatures elevate the development and growth rates which shape the body size, here reflected as body mass. Ectotherms that follow the reverse temperature size rule (TSR) phenomenon, as found in our case, are expected to be more temperature-sensitive for biomass accumulation rather than for ontogenetic development [77]. Similar results in which body size increased with increased temperature were documented in some odonates [78], amphibians [79], and fishes [80, 81]. An increased size under warming could have a positive impact on fecundity; usually, larger females have higher potential fecundity than smaller ones [82], but see [83]. In contrast, other

studies on *I. elegans* showed no or the opposite pattern where elevated temperatures decreased mass at emergence [13, 29, 32, 57]. In these cases, larvae were exposed to strong intraspecific competition which might cause the discrepancy in the results. Previous and current findings suggest that it is not straightforward to predict future changes in ectotherms' size under warming conditions using simple experimental studies.

When comparing between populations, the adaptive response to warming and predation stress with faster completion of pre-emergence development in the Płaszowski compared to the Dąbski population was traded off with a lower mass at emergence. Therefore, fixed population-specific differences in mass with lighter individuals from the warmer than the colder pond opposed the plastic thermal responses expressed during the experiment by both populations. This suggests that short-term ecological responses to warming might result in an increased mass, whereas long-term evolutionary responses in mass gain might be constrained by co-gradient thermal variation in an associated trait, development rate. As mentioned, maternal effects that were not analyzed here could also play a role in shaping mass. This makes it challenging to conclude to what degree the direction of thermal response in mass is driven by plasticity, genes, or maternal effects.

## Conclusions

Our results revealed the presence of population- and life-stage-specific responses in life-history traits to NCEs from native and invasive predators and warming. These population-specific responses to ecological stressors were recorded at a microgeographic scale, suggesting a phenotypic signal of adaptation despite a likely strong gene flow, as shown in a recent study on the study species [76]. The detected micro-geographic differences in the responses to warming and predator cues may inform why biocontrol strategies using predators may fail to control pests like mosquitoes at particular breeding sites but succeed at other sites. Such biological control implications were discussed earlier in many studies [84, 85]. Finally, we observed pronounced shifts during ontogeny from synergistic (egg and early larval stage) toward additive (late larval stage up to emergence) effects between warming and predator stress. Stage-specific responses to (combined) stressors highlight that a full life cycle approach is required to predict adult fitness through carry-over effects of stressors experienced in egg and larval stages.

## Supporting information

**S1 Fig. Temperature and photoperiod regimes used during the experiment.** Panel (A) shows mean temperature regimes under the current temperature and +4˚C warming treatments modeled using FLake modeling (Lake Model FLake. 2009). Panel (B) represents mean photoperiod regimes. The x-axis in both panels represents the timeline of the experiment, from egg hatching (week 0) until the last larva emerged (week 46).
(TIF)

**S2 Fig. Average daily temperature from 1 March to 2 June 2023 in Dąbski and Płaszowski ponds which are located in the city of Kraków, Poland.**
(TIF)

**S3 Fig. Effects of sex on the estimated mean mass at emergence in *I. elegans* from two populations in Dąbski and Płaszowski ponds.** Error bars show 95% CI.
(TIF)

**S1 Table. The values of photoperiod, temperatures, and dates for which current values were taken.** In the experiment, the overwintering was shortened compared to nature. Therefore, during post-winter treatment, the dates for experimental values did not follow the current

values in nature at a particular time point.
(DOCX)

**S2 Table. Sample sizes across the experimental groups.** Note that in cases when group sample size < 2, these groups were excluded from the analysis.
(DOCX)

**S3 Table. Sample sizes of accidentally lost individuals across the experimental groups.**
(DOCX)

**S4 Table. Study ponds characteristics.** Dąbski and Płaszowski ponds are located in the city of Kraków, Poland. Both ponds represent the same type of freshwater habitat, similar age, and the same original and current purpose. The distance between the ponds is 2,8 km, and both ponds are situated within the Wisła river valley.
(DOCX)

**S5 Table. Average daily temperature differentiation between Dąbski and Płaszowski ponds from 1 March to 2 June 2023.** Data were extracted from five loggers installed in the two ponds (three loggers in Płaszowski pond and two loggers in Dąbski pond) ca. 40 cm below the water surface. Significant p-values are in bold.
(DOCX)

## Acknowledgments

We would like to thank E. Sieńko, M. Nowicki, and P. Laskowski for help in crayfish catching, M. Bonk and R. Maciaszek for information on crayfish populations and M. Raczyński for support in field and laboratory work. Comments from two reviewers considerably improved the manuscript.

## Author Contributions

**Conceptualization:** Andrzej Antoł, Szymon Sniegula.

**Formal analysis:** Nermeen R. Amer, Andrzej Antoł.

**Investigation:** Szymon Sniegula.

**Methodology:** Andrzej Antoł, Szymon Sniegula.

**Supervision:** Szymon Sniegula.

**Visualization:** Nermeen R. Amer.

**Writing – original draft:** Nermeen R. Amer.

**Writing – review & editing:** Nermeen R. Amer, Robby Stoks, Szymon Sniegula.

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
