## [Decision Letter · Decision Letter 0]

8 Oct 2023

PONE-D-23-24816­­Microgeographic differentiation in thermal and antipredator responses and their carry-over effects across life stages in a damselflyPLOS ONE

Dear Dr. Amer,

Thank you for submitting your manuscript to PLOS ONE.  We have been able to get comments from the reviewers. After careful consideration, we feel that it has merit but does not fully meet PLOS ONE’s publication criteria as it currently stands. Therefore, we invite you to submit a revised version of the manuscript that addresses the points raised during the review process.I personally realize that the results obtained herein e.g. decreased warming extended larval developmental duration by predator with negative carry-over effect from egg to larva will have long  term implication on combined effect of warming and invasion of alien species. The results will have implications on global warming impacts on biological control of pests and vectors using predators. Results might answer the question like why biocontrol agents fail to control mosquitoes at particular breeding site but succeed at other sites. I would recommend that in addition to fitness consequences authors should bring these issues in discussion section. Please ensure that necessary ethical clearances have been taken for this study.   Please submit your revised manuscript by Nov 22 2023 11:59PM. If you will need more time than this to complete your revisions, please reply to this message or contact the journal office at plosone@plos.org. Please include the following items when submitting your revised manuscript:A rebuttal letter that responds to each point raised by the academic editor and reviewer(s). You should upload this letter as a separate file labeled 'Response to Reviewers'.A marked-up copy of your manuscript that highlights changes made to the original version. You should upload this as a separate file labeled 'Revised Manuscript with Track Changes'.An unmarked version of your revised paper without tracked changes. You should upload this as a separate file labeled 'Manuscript'.If applicable, we recommend that you deposit your laboratory protocols in protocols.io to enhance the reproducibility of your results. Protocols.io assigns your protocol its own identifier (DOI) so that it can be cited independently in the future. For instructions see: https://journals.plos.org/plosone/s/submission-guidelines#loc-laboratory-protocols. Additionally, PLOS ONE offers an option for publishing peer-reviewed Lab Protocol articles, which describe protocols hosted on protocols.io. Read more information on sharing protocols at https://plos.org/protocols?utm_medium=editorial-email&utm_source=authorletters&utm_campaign=protocols.

We look forward to receiving your revised manuscript.

Kind regards,

Ram Kumar, Ph.D.

Academic Editor

PLOS ONE

Journal Requirements:

Additional Editor Comments:

The results have potential to be published, however the application of this study can be broaden like impacts of inclusive fitness, biocontrol of disease vectors by predators, Differential efficiency of different predators to control vectors at different micrgeographical habitats etc.

The results obtained herein e.g. decreased warming extended larval developmental duration by predator with negative carry-over effect from egg to larva will have long term implication on combined effect of warming and invasion of alien species. The results will have implications on global warming impacts on biological control of pests and vectors using predators. Results might answer the question like why biocontrol agents fail to control mosquitoes at particular breeding site but succeed at other sites. I would recommend that in addition to fitness consequences authors should bring these issues in discussion section.

Following references might be useful in explain further consequences predator deterrent on life stages and different populations

1. Effect of Mesocyclops thermocyclopoides (Copepoda, Cyclopoida) predation on population dynamics of different prey: a laboratory study; January 2003; Journal of Freshwater Ecology 18(3):383 – 393

• 2. Predation on Mosquito Larvae by Mesocyclops thermocyclopoides (Copepoda: Cyclopoida) in the Presence of Alternate Prey. November 2003, Internationale Revue der gesamten Hydrobiologie und Hydrographie, 88(6):570 – 581DOI: 10.1002/iroh.200310631

3. How effective are Mesocyclops aspericornis (Copepoda: Cyclopoida) in controlling mosquito immatures in the environment with an application of phytochemicals?October 2013; Hydrobiologia 716(1). DOI:10.1007/s10750-013-1559-9

4. Biological mosquito control is affected by alternative prey; July 2015 ; Zoological Studies 54:55(1). DOI:: 10.1186/s40555-015-0132-9

5. Potential of three aquatic predators to control mosquitoes in the presence of alternative prey: A comparative experimental assessment; January 2008; Marine and Freshwater Research 59(9). DOI: 10.1071/MF07143

6. Impacts of predation by the copepod, Mesocyclops pehpeiensis, on life table demographics and population dynamics of four cladoceran species: A comparative laboratory study; November 2009; Zoological Studies 48(6):738-752

Reviewers' comments:

Reviewer's Responses to Questions

**Comments to the Author**

1. Is the manuscript technically sound, and do the data support the conclusions?

Reviewer #1: Yes

Reviewer #2: Yes

2. Has the statistical analysis been performed appropriately and rigorously? 

Reviewer #1: Yes

Reviewer #2: Yes

3. Have the authors made all data underlying the findings in their manuscript fully available?

Reviewer #1: Yes

Reviewer #2: Yes

4. Is the manuscript presented in an intelligible fashion and written in standard English?

Reviewer #1: Yes

Reviewer #2: Yes

5. Review Comments to the Author

Reviewer #1: The manuscript dealt with the impact of global warming and the predation of invasive species against the native species of damselfly, Ischnura elegans, across egg and larval stages. This kind of work will certainly be of interest to the readers. However, the manuscript requires further minor clarification that need to be addressed by the authors.

1. What is the selection reason of the damselfly Ischnura elegans for the study? Highlight its ecological importance in the manuscript.

2. How many replications have been used in the experiments? Mention the details in the table.

3. There are similar kinds of studies available with I. elegans. Highlight the novelty of your research in the manuscript.

4. If the conclusion section extracted from the discussion would be better.

5. The entire manuscript requires a minor grammatical check.

Other small mistakes

1. Line number 180 : My Worms should be written as myWORMS or MYWORMS

2. Line number 175 : From

3. Unnecessary spaces between words ( line no. 196,212)

4. Line no.110: 22 °C, Line no. 129: 4 °C, line no. 134: 6 °C, line no. 135: 10 °C but in Line no. 174,193,196 it is written as 20°C, 10°C, 12°C respectively

5. Line no. 387: cogradient should be Co-Gradient

6. Line no. 412: xxx what?xxx

7. Line no. 359: and/or

8. Line no. 105: copula

9. Line no. 173: 52 L ( gap)

10. Line no. 177: 100 g, line no. 178: 41.5 g, line no. 181: 10 L ( gap)

11. Line no 484 : Ades should be written in italics

12. Line no. 513, 528, 531,538,: I. elegans should be in italics

13. Line no 542: species name in italics

14. Line no. 566: species name in italics

15. Line no 568,569 species name in italics

16. Line no 593 species name in italics

17. Line no 596 species name in italics

18. Line no 616 species name in italics

19. Line no 618 species name in italics

20. Line no 619: Http – http

21. Line no 632,633 species name in italics

22. Line no 657 species name in italics

23. Figure should be in Fig. form

24. KEY WORD should be KEYWORD

25. Acknowledgment is missing

26. They can give conclusion separately if they want

Reviewer #2: The manuscript is well written and provides detail insight on the possible combined impact of global warming and the invasion of non-native predator species on the egg and larval behavior of the very common damselfly. This experimental work gives us new understanding about the overall prey population behaviors and also specific adaptive behavior at different life stages of prey species. Knowing the importance of the organism in the trophic structure and energy transfer and the information on how climate induced changes may lead to a cascading effect (top-down or bottom-up) is the need of hour. I would especially like to mention that, this research has adopted an experimental approach to address the combined effects of temperature rise and predation (native and non-native) effects on the life cycle of prey population. Though the experimental findings are difficult to acclimatize into real world environment, a setup of multiple test combinations have answered many questions and based on this observation the authors were able to demonstrate that, the ecologically important damselfly Ischnura elegans exhibits adaptive resilience to warming and predator stress.

I believe under ever-changing climate and ever-increasing human pressure on the world global ecosystem, this manuscript will fall in the line of global efforts of ecosystem conservation and future prediction model. This work also provided a promising finding which is quite opposite to the prevailing notion on the adverse impact due to the climate change which must be supported and substantiated later. Therefore, I recommend the manuscript has a potential for publication in PLOSE ONE. However, few of minor corrections from my side have been highlighted, corrected or deleted in the PDF file and request the authors to kindly incorporate the same as long as those are empirical. Since I am not a native of English speaking but still, I gave a try to correct few.

6. PLOS authors have the option to publish the peer review history of their article (what does this mean?). If published, this will include your full peer review and any attached files.

Reviewer #1: **Yes: **Dr Jatin Kalita

Reviewer #2: **Yes: **Jawed Equbal

---

## [Author Response · Author response to Decision Letter 0]

13 Nov 2023

Reviewer #1: 

The manuscript dealt with the impact of global warming and the predation of invasive species against the native species of damselfly, Ischnura elegans, across egg and larval stages. This kind of work will certainly be of interest to the readers. However, the manuscript requires further minor clarification that need to be addressed by the authors.

1. What is the selection reason of the damselfly Ischnura elegans for the study? Highlight its ecological importance in the manuscript.

Thank you for this comment. We rephrased this paragraph and added these highlighted sentences.

We chose I. elegans because it is a very common damselfly occurring from mid-Scandinavia and the United Kingdom to southern Italy and southern Spain [33], and it is neither protected nor endangered. In central Europe, the damselfly has one or two generations per year [34] with diapausing larvae as the overwintering stage [35]. Larvae hatch a couple of weeks after egg laying. Aquatic egg and larval stages commonly share habitats with top predators such as fish and crayfish [17,36–38]. Moreover, I. elegans larvae occupy an intermediate position in aquatic food webs. They are significant predators of many aquatic invertebrates, including midges and mosquito larvae, while being prey for larger predatory invertebrates and fish [18]. It was shown that predator cues affect I. elegans damselfly life history and physiology during both the egg stage and larval stage [13,17,39]. Lines (78-85) 

2. How many replications have been used in the experiments? Mention the details in the table.

Table S2 and S3 include the sample sizes and the number of accidentally lost individuals respectively (Supplementary material file). 

In the manuscript, we added “At hatching, larvae were transferred to other containers (19 x 12 x 9 cm) filled with 1 l of water and kept in groups of 15 individuals (two group containers per treatment, a total of 30 individuals per treatment) for another 14 days. Lines (161-162).

3. There are similar kinds of studies available with I. elegans. Highlight the novelty of your research in the manuscript.

We added “ While similar studies on I. elegans exist [29,32], our current research is novel because it examines the interactive effects of two important ecological stressors, global warming and biological invasion, throughout various life stages in prey (egg and larva). Additionally, our study focuses on different response patterns between populations at a micro-geographical scale.” (Lines 63-67)

4. If the conclusion section extracted from the discussion would be better.

We separated it (Lines 430-442)

5. The entire manuscript requires a minor grammatical check.

Thank you, we checked it in the entire manuscript (highlighted)

“Reflects” (line 198)

“a wet” (line 203)

“experiences” (line 341)

“is” (line 395)

“a”, “the” (line 396)

“the” (line 397)

“conditions” (line 419)

We changed “defence” into “defense” (lines 34,333, 347)

Other small mistakes

1. Line number 180 : My Worms should be written as myWORMS or MYWORMS 

Done, we changed it into “myWORMS” (line 187)

2. Line number 175 : From

Thank you, but in the sentence, “from” is a preposition, not a noun, I assume it should be “from” as it is. (line 182)

3. Unnecessary spaces between words ( line no. 196,212)

Removed.

4. Line no.110: 22 °C, Line no. 129: 4 °C, line no. 134: 6 °C, line no. 135: 10 °C but in Line no. 174,193,196 it is written as 20°C, 10°C, 12°C respectively

We adjusted it according to PLOS One formats. Now, all temperature degrees are in this format “6°C” (highlighted throughout the manuscript). Lines (116,121,135,140,141,181,200,245,268,445)

5. Line no. 387: cogradient should be Co-Gradient

Modified (line 396,426)

6. Line no. 412: xxx what?xxx

We added “the adaptive response to warming and predation stress” (line 420)

7. Line no. 359: and/or

We added a space and/or (line 368)

8. Line no. 105: copula

Modified to “copulation” (line 111)

9. Line no. 173: 52 L ( gap)

According to PLOS One formats, it should be a space between numbers and units. Modified and highlighted throughout the manuscript. Lines (180, 184, 185,188)

10. Line no. 177: 100 g, line no. 178: 41.5 g, line no. 181: 10 L ( gap)

We followed PLOS One format between numbers and units (e.g., line 184:100 g, line 185: 41.5 g, line 188: 10 L)

11. Line no 484 : Ades should be written in italics

Done (line 526)

12. Line no. 513, 528, 531,538,: I. elegans should be in italics

Done (lines 555, 573, 576)

13. Line no 542: species name in italics

Done (line 583)

14. Line no. 566: species name in italics

Done (line 587)

15. Line no 568,569 species name in italics

Done (lines 612, 614)

16. Line no 593 species name in italics

Done (line 639)

17. Line no 596 species name in italics

Done (line 642)

18. Line no 616 species name in italics

Done (line 662)

19. Line no 618 species name in italics

Done (line 664)

20. Line no 619: Http – http

Done (line 665)

21. Line no 632,633 species name in italics

Done (line 678,679,685)

22. Line no 657 species name in italics

Done (line 706)

23. Figure should be in Fig. form

Done (lines95,122,126,127,142,145,233,243,245,252,254,256,268,279,282,285,288,294,299,301,303,376,444,449,451)

24. KEY WORD should be KEYWORD

Done (line 26)

25. Acknowledgment is missing

We added the Acknowledgment section (Lines 469-473)

26. They can give conclusion separately if they want

We separated the conclusion section (Lines 430-442).

Reviewer #2: 

The manuscript is well written and provides detail insight on the possible combined impact of global warming and the invasion of non-native predator species on the egg and larval behavior of the very common damselfly. This experimental work gives us new understanding about the overall prey population behaviors and also specific adaptive behavior at different life stages of prey species. Knowing the importance of the organism in the trophic structure and energy transfer and the information on how climate induced changes may lead to a cascading effect (top-down or bottom-up) is the need of hour. I would especially like to mention that, this research has adopted an experimental approach to address the combined effects of temperature rise and predation (native and non-native) effects on the life cycle of prey population. Though the experimental findings are difficult to acclimatize into real world environment, a setup of multiple test combinations have answered many questions and based on this observation the authors were able to demonstrate that, the ecologically important damselfly Ischnura elegans exhibits adaptive resilience to warming and predator stress.

I believe under ever-changing climate and ever-increasing human pressure on the world global ecosystem, this manuscript will fall in the line of global efforts of ecosystem conservation and future prediction model. This work also provided a promising finding which is quite opposite to the prevailing notion on the adverse impact due to the climate change which must be supported and substantiated later. Therefore, I recommend the manuscript has a potential for publication in PLOSE ONE. However, few of minor corrections from my side have been highlighted, corrected or deleted in the PDF file and request the authors to kindly incorporate the same as long as those are empirical. Since I am not a native of English speaking but still, I gave a try to correct few.

Line 8: This is more technical (Non- consumptive predator effects Cant it be more general?

We changed it into “predator stress” (Line 26)

Line 11: Rephrase or break the sentence. Global warming and Invasion of exotic species are two different phenomena. Global warming could be the sole reason of invasion but together both the phenomenon will have two different effects or might be effecting singularly on the native species. 

We changed it into” Global warming and invasive species, separately or combined, can impose a large impact on the condition of native species” (Line 2)

Line 20: Survival for what? If you mean damselfly? Then kindly mention " survival of damselfly"

We added “damselfly” survival (Line 10)

Line 32-33: The first sentence of the paragraph should not start with the helping articles like as, of, in, to, which can mostly be avoided unless necessary. Kindly rephrase

We rephrased it into “Organisms are typically exposed to multiple stressors which may interact so, it is important to study the individual and combined effects of these stressors to accurately assess their impact [1–4]” (Line 29-30)

Line 34: "activities"

We changed it into “activities” (Line 31)

Line 56: what is "salamander chemical cues" this can be elaborated?

We changed it into “salamander kairomones and mosquito alarm cues” (Line 53)

Line 72: What does mean of "mass" kindly elaborate once and carry over.

We changed it into “body mass” (Line 73)

Line 76: Kindly rephrase the sub heading. "Target species and the study area" or as such like that?

We changed it into “Target species and study area” (Line 77)

Line 89: “our” Why it will be yours, Remove this

Removed

Additional Editor Comments:

The results have potential to be published, however the application of this study can be broaden like impacts of inclusive fitness, biocontrol of disease vectors by predators, Differential efficiency of different predators to control vectors at different micrgeographical habitats etc.

The results obtained herein e.g. decreased warming extended larval developmental duration by predator with negative carry-over effect from egg to larva will have long term implication on combined effect of warming and invasion of alien species. The results will have implications on global warming impacts on biological control of pests and vectors using predators. Results might answer the question like why biocontrol agents fail to control mosquitoes at particular breeding site but succeed at other sites. I would recommend that in addition to fitness consequences authors should bring these issues in discussion section.

Following references might be useful in explain further consequences predator deterrent on life stages and different populations

1. Effect of Mesocyclops thermocyclopoides (Copepoda, Cyclopoida) predation on population dynamics of different prey: a laboratory study; January 2003; Journal of Freshwater Ecology 18(3):383 – 393

2. Predation on Mosquito Larvae by Mesocyclops thermocyclopoides (Copepoda: Cyclopoida) in the Presence of Alternate Prey. November 2003, Internationale Revue der gesamten Hydrobiologie und Hydrographie, 88(6):570 – 581DOI: 10.1002/iroh.200310631

3. How effective are Mesocyclops aspericornis (Copepoda: Cyclopoida) in controlling mosquito immatures in the environment with an application of phytochemicals?October 2013; Hydrobiologia 716(1). DOI:10.1007/s10750-013-1559-9

4. Biological mosquito control is affected by alternative prey; July 2015 ; Zoological Studies 54:55(1). DOI:: 10.1186/s40555-015-0132-9

5. Potential of three aquatic predators to control mosquitoes in the presence of alternative prey: A comparative experimental assessment; January 2008; Marine and Freshwater Research 59(9). DOI: 10.1071/MF07143

6. Impacts of predation by the copepod, Mesocyclops pehpeiensis, on life table demographics and population dynamics of four cladoceran species: A comparative laboratory study; November 2009; Zoological Studies 48(6):738-752

We agree this could be an interesting angle for the discussion, and now added a discussion part on how our results may inform about the implications of global warming on the biological control of pests and vectors using predators. Yet, to avoid being too speculative we kept it short. We added

“The detected micro-geographic differences in the responses to warming and predator cues may inform why biocontrol strategies using predators may fail to control pests like mosquitoes at particular breeding sites but succeed at other sites. Such biological control implications were discussed earlier in many studies [84,85]” (Lines 435-438)

Other author modifications:

1- We added supporting information (lines 443-468)

2- We added 4 new references no (47,76, 84,85), modified two references (32,55), and deleted one reference (44).

3- We noticed and modified a small mistake in the total sample size of “survival one day after emergence (Table S2).

4- Due to the changes in survival data, we modified Fig.3-B and made small changes in the abstract section (highlighted- lines 9-13), the result section (lines 262-266), and the discussion part (modified survival paragraph lines 314-320). 

Specifically, our original analysis of survival until emergence was based on the survival between day 14 after hatching and emergence. The updated analysis of survival until emergence is based on the total time between hatching and emergence. We want to note that the updated results do not differ qualitatively from the original ones. Therefore, only minor changes in the text and Fig.3-B were introduced.

5- We added one more sentence in the abstract section “We also observed pronounced shifts during ontogeny from synergistic (egg and early larval stage) toward additive (late larval stage up to emergence) effects between warming and predator stress.” (Lines 21-23)

6- On the submission website, we want to add these three funding resources:

1. Narodowe Centrum Nauki, Grant/Award Number: 2019/33/B/NZ8/00521

2. Norwegian Financial Mechanism 2014-2021, Grant/Award Number: 2019/34/H/NZ8/00683 (ECOPOND)

3. Institute of Nature Conservation Polish Academy of Sciences

---

## [Decision Letter · Decision Letter 1]

27 Nov 2023

­­Microgeographic differentiation in thermal and antipredator responses and their carry-over effects across life stages in a damselfly

PONE-D-23-24816R1

Dear Dr. Amer,

We’re pleased to inform you that your manuscript has been judged scientifically suitable for publication and will be formally accepted for publication once it meets all outstanding technical requirements. Thank you for considering PloseOne for dissemination of your research.   

Kind regards,

Ram Kumar, Ph.D.

Academic Editor

PLOS ONE

Additional Editor Comments (optional):

Reviewers' comments:

Reviewer's Responses to Questions

**Comments to the Author**

1. If the authors have adequately addressed your comments raised in a previous round of review and you feel that this manuscript is now acceptable for publication, you may indicate that here to bypass the “Comments to the Author” section, enter your conflict of interest statement in the “Confidential to Editor” section, and submit your "Accept" recommendation.

Reviewer #2: All comments have been addressed

2. Is the manuscript technically sound, and do the data support the conclusions?

Reviewer #2: Yes

3. Has the statistical analysis been performed appropriately and rigorously? 

Reviewer #2: Yes

4. Have the authors made all data underlying the findings in their manuscript fully available?

Reviewer #2: Yes

5. Is the manuscript presented in an intelligible fashion and written in standard English?

Reviewer #2: Yes

6. Review Comments to the Author

Reviewer #2: The comments from the reviewer side has been incorporated. The manuscript is well written and provides detail insight on the possible combined impact of global warming and the invasion of non-native predator species on the egg and larval behavior of the very common damselfly. This experimental work gives us new understanding about the overall prey population behaviors and also specific adaptive behavior at different life stages of prey species. Knowing the importance of the organism in the trophic structure and energy transfer and the information on how climate induced changes may lead to a cascading effect (top-down or bottom-up) is the need of hour. I would especially like to mention that, this research has adopted an experimental approach to address the combined effects of temperature rise and predation (native and non-native) effects on the life cycle of prey population. Though the experimental findings are difficult to acclimatize into real world environment, a setup of multiple test combinations have answered many questions and based on this observation the authors were able to demonstrate that, the ecologically important damselfly Ischnura elegans exhibits adaptive resilience to warming and predator stress.

I believe under ever-changing climate and ever-increasing human pressure on the world global ecosystem, this manuscript will fall in the line of global efforts of ecosystem conservation and future prediction model. This work also provided a promising finding which is quite opposite to the prevailing notion on the adverse impact due to the climate change which must be supported and substantiated later. Therefore, I recommend the manuscript has a potential for publication in PLOSE ONE. However, few of minor corrections from my side have been highlighted, corrected or deleted in the PDF file and request the authors to kindly incorporate the same as long as those are empirical. Since I am not a native of English speaking but still, I gave a try to correct few.

7. PLOS authors have the option to publish the peer review history of their article (what does this mean?). If published, this will include your full peer review and any attached files.

Reviewer #2: **Yes: **JAWED EQUBAL

---

## [Editor Report · Acceptance letter]

7 Dec 2023

PONE-D-23-24816R1 

­­Microgeographic differentiation in thermal and antipredator responses and their carry-over effects across life stages in a damselfly 

Dear Dr. Amer:

I'm pleased to inform you that your manuscript has been deemed suitable for publication in PLOS ONE. Congratulations! Your manuscript is now with our production department. 

Kind regards, 

on behalf of

Professor Ram Kumar 

Academic Editor

PLOS ONE